# The Preparation and Biological Testing of Novel Wound Dressings with an Encapsulated Antibacterial and Antioxidant Substance

**DOI:** 10.3390/nano12213824

**Published:** 2022-10-29

**Authors:** Petr Braťka, Taťána Fenclová, Jana Hlinková, Lenka Uherková, Eva Šebová, Veronika Hefka Blahnová, Věra Hedvičáková, Radmila Žižková, Andrej Litvinec, Tomáš Trč, Jozef Rosina, Eva Filová

**Affiliations:** 1Faculty of Biomedical Engineering, Czech Technical University, Náměstí Sítná 3105, 27201 Kladno, Czech Republic; 2Grade Medical s.r.o., Náměstí Sítná 3105, 27201 Kladno, Czech Republic; 3Department of Tissue Engineering, Institute of Experimental Medicine of the Czech Academy of Sciences, Vídeňská 1083, 14220 Prague, Czech Republic

**Keywords:** wound dressing, antibacterial, antioxidant, encapsulation, nanofiber substrate, personalized care, in vivo

## Abstract

Chronic wounds represent a significant socio-economic problem, and the improvement of their healing is therefore an essential issue. This paper describes the preparation and biological properties of a novel functionalized nanofiber wound dressing consisting of a polycaprolactone nanofiber carrier modified by a drug delivery system, based on the lipid particles formed by 1-tetradecanol and encapsulated gentamicin and tocopherol acetate. The cytotoxicity of extracts was tested using a metabolic activity assay, and the antibacterial properties of the extracts were tested in vitro on the bacterial strains *Staphylococcus aureus* and *Pseudomonas aeruginosa*. The effect of the wound dressing on chronic wound healing was subsequently tested using a mouse model. Fourteen days after surgery, the groups treated by the examined wound cover showed a lower granulation, reepithelization, and inflammation score compared to both the uninfected groups, a lower dermis organization compared to the control, a higher scar thickness compared to the other groups, and a higher thickness of hypodermis and bacteria score compared to both the uninfected groups. This work demonstrates the basic parameters of the safety (biocompatibility) and performance (effect on healing) of the dressing as a medical device and indicates the feasibility of the concept of its preparation in outpatient conditions using a suitable functionalization device.

## 1. Introduction

Wound care management is growing due to the increase in the world’s population and the aging population. The treatment of skin defects, especially chronic wounds, is an important topic since the skin serves as the main protection of the human body against pollution from the surrounding area. A defect in the skin is usually caused by mechanical, thermal, chemical, or pathological conditions, which influence the wound area and the surrounding tissue. These conditions affect the healing process of the wound, which then differs depending on its type. The main groups are acute and chronic wounds, which particularly differ during the healing process. Acute wounds heal in less than 3 months, while chronic wounds take more than 3 months to heal or can remain in a chronic phase [1].

In view of the above, we designed, prepared, and tested the biological properties of a nanofiber wound dressing, functionalized with antibacterial and antioxidative substances, intended for the treatment of chronic wounds. The fundamental benefit of this wound dressing is shown in the improved wound care given by the prolonged release of active substances that provide a combined or rather synergic effect on the one hand, and by the unique properties of the nanofibrous carrier on the other hand. Another significant advantage is that the designed preparation procedure allows a certain modification of the parameters of each single piece of wound cover, according to the individual needs of the patient and, therefore, also personalized care.

The chronic phase of the wound is affected by the patient’s other potential illnesses, e.g., diabetes. In particular, comorbidity patients have a high probability of the occurrence of bacterial infected non-healing wounds. Therefore, the complementary antimicrobial effect of the wound dressing is essential.

Gentamicin sulphate is an aminoglycoside molecule that exhibits a bactericidal activity against a broad spectrum of microorganisms, such as *Pseudomonas aeruginosa (P. aeruginosa)*, *Escherichia coli,* and *Staphylococcus aureus (S. aureus)*, and many other Gram-positive and Gram-negative bacteria strains [2,3]. Gentamicin sulphate is already used in topical applications and it seems suitable for use as an antimicrobial agent effective against the bacteria which are common in chronic wound bacterial infections.

Alternatively, the chemical composition of the wound environment, especially of wounds associated with diabetes, could significantly influence cell proliferation, the difficulty of bacterial infection treatment, and the physiological and structural changes in the wound [4]. Many research articles show that, for example, a low level of reactive oxygen species (ROS) helps towards a normal healing of the wound, normal cell migration and angiogenesis, and, on the contrary, a high concentration leads to oxidative stress and can significantly worsen wound healing, especially in the case of chronic wounds. A long- lasting inflammatory response in chronic wounds leads to a large accumulation of ROS, which exceeds the antioxidant capacity of the cells. Wounds cannot transit from the inflammatory phase to the proliferative phase. To solve this state of the wound, many substances aiming to remove ROS from the wound are studied, offering antioxidants as the most functional solution. Antioxidants can decrease the concentration of free radicals in wounds [5]. The combination of ROS concentration decreases while treating the bacterial infection; using antibiotics in wound dressings can change the environment of the chronic wound and can lead to the proliferation phase.

Reactive oxygen species are basically free radicals, so are reactive chemical substances with an unpaired electron in an outer orbit. They are produced continuously when cells create energy from metabolic reactions and oxygen and, to an increased extent, when exposed to microbial infections, an extensive effort, or pollutants/toxins such as cigarette smoke, alcohol, ionizing and UV radiations, pesticides, and the ozone. Antioxidants are defined as substances that decrease the levels of free radicals including, ROS, or prevent its production. For the good proliferation of cells, it is necessary to obtain the correct balance between oxidative and antioxidative forces. It is also necessary to find a balance between the concentration of ROS and antioxidants, to get the best environment for cell proliferation and the related wound healing [6]. Therefore, the prolonged release of an antioxidative agent was also determined as an important tissue of the designed dressing.

Tocopherol acetate belongs to the group of naturally occurring antioxidants listed commonly under the name vitamin E (VitE). It is widely used in the cosmetic and food supplement industry. However, it is more important that it has a proven antioxidative effect in the topical environment [7]; although the effectiveness of its forms is often discussed. Since tocopherol acetate seemed, based on its known chemical properties, compatible with the gentamicin, it was chosen as the antioxidative agent.

Nanofibers and nanofibrous nonwovens produced of polycaprolactone (PCL) have proven to be a suitable carrier for use in human medicine, especially for the preparation of sophisticated wound covers or scaffolds modified by deposited lipid-based drug delivery systems with a proven biocompatibility [8]. The morphology of nanofiber substrates provides, in addition to the high adsorption capacity for lipid particles, a suitable combination of properties for the healing of chronic wounds (exudate drainage, water vapor permeability, and barrier function against bacteria contamination) depending on the specific design.

There are many new biocompatible materials that deliver an antibacterial activity which is efficient against pathogenic bacterial strains. There are different structures, compositions, and methods of preparation of these materials; some are even effective against antibiotic-resistant strains. Silver nanoparticles present one type of these biomaterials. They are being tested not only on wound dressings, but also on other materials with the aim of preventing skin infections [9]. Nanostructures with loaded active substances represent another type of novel materials. Ceylan et al. prepared a gentamicin-loaded PCL nanofiber, where the higher surface to volume ratio offers a larger contact surface with wound and hydrophobic surfaces; this slows down the release rate of the incorporated drug [10]. This common novel approach creates antibiotic carriers with antibiotics incorporated in the lipid structures forming liposomes or solid lipid particles, where the lipid helps to increase the antibacterial efficiency at a lower drug concentration. Mugabe et al. compared the antibacterial effect of a free drug and a drug encapsulated in liposomes made of 1,2-Dipalmitoyl-sn-glycero-3-phosphocholine and cholesterol. In their study, they described a nonmucoid bacterial strain which was highly resistant to free amikacin (minimum inhibitory concentration [MIC] of 256 μg/mL) and tobramycin (MIC, 64 μg/mL), but highly sensitive (MIC, ≤8 μg/mL) to the aforementioned antibiotics encapsulated in the liposomes. The combination of free liposomal particles with a free drug showed no additive effect on the antibacterial activity [11]. The efficiency of tetracycline on a tetracycline-resistant bacteria strain after its encapsulation was shown by Ghosh et al. in their study [12]. Rucholm et al.’s study shows that the MIC of 32 mg/L for *P. aeruginosa* is significantly lower for liposomal gentamicin than the MIC of 512 mg/L for free gentamicin. The liposomal loading of antibiotics allows the appliance of lower doses of antibiotics to prevent the antibiotic resistance of bacterial strains [13].

Another essential element that significantly contributes to better results of the treatment process is the personalization of wound care [14]. The clinical use of a personalized wound cover, i.e., a cover whose parameters are at least to some extent modified according to the needs of a specific patient, is more or less only realistic using an ad hoc preparation of the cover in ambulatory conditions. Therefore, the primary search was for a method of preparing the formulation and depositing it on the carrier, which can be prepared in outpatient conditions, and in particular using a functionalization device of the company’s own design, Grade Medical s.r.o. (limited liability company). In summary, the aim of this study was to prepare a novel wound dressing with an antibacterial activity against the bacterial strains typical of chronic wounds and, at the same time, with an antioxidant activity to decrease the oxidative stress in chronic wounds showing further advantageous properties of the carrier, such as a good gas exchange; this demonstrates the basic parameters of safety (biocompatibility) and performance (contribution to the wound healing) and the basic suitability of the concept for personalized wound care.

## 2. Materials and Methods

### 2.1. Materials

Polycaprolactone (Mw = 45,000, Sigma Aldrich, Saint Luis, MO, USA) was used for the nanofiber substrates and acetic acid (99% pure, Penta Chemicals, Prague, Czech Republic), chloroform (VWR Chemicals, Leuven, Belgium), ethanol (96%, P-lab, Prague, Czech Republic), and acetone (AC) (P-lab/Carl Roth GmbH, Karlsruhe, Germany) were used as the solvents. Absolute ethanol (Penta Chemicals, Prague, Czech Republic) was used for washing the fibers after their preparation and for disinfection. The solid lipid particles (SLP) were prepared from 1-tetradecanol (Sigma Aldrich, Steinheim, Germany), distilled water, Tween 20 (Molecular biology grade, WVR Chemicals, Solon, OH, USA), and ethanol (absolute, Penta Chemicals in, Prague, Czech Republic) an antioxidant SLP preparation. Gentamicin sulphate (BioChemica, Darmstadt, Germany) was used as an antibacterial substance in a solution in distilled water. α-tocopherol was used as the antioxidant substance (from vegetable oil, type V, Sigma Aldrich, Saint Luis, MO, USA).

### 2.2. Equipment

The structures of the materials were evaluated by the scanning electron microscopy (SEM, Phenom Pro, Eindhoven, The Netherlands), the bacterial concentration was measured on a densitometer (McFarland, DEN-1, Cambs, England), the particle size distribution on Mastersize 2000 (Malvern, Worcestershire, United Kingdom), and the release of the active substance was measured on HPLC-FLD (Agilent HPLC 1200, Hewlett-Packard, Waldbronn, Germany). The nanofiber substrates were prepared on electrospinning machines. The absorbance was measured by a spectrophotometer Tecan Infinite of 200 M (Tecan, Grodig, Austria).

For the synthesis and deposition of the formulations on the carriers, experimental, semi-operational equipment of Grade Medical’s own design was used. This device, with a variable configuration, initially consists of precision linear dispensers, magnetic stirrers, microfluidic chips (not used), and a deposition chamber to place a cartridge with a substrate with the possibility of vacuum deposition (not used).

### 2.3. Nanofiber Substrate Preparation Method

The PCL fibers were prepared by electrospinning using two machines with edge electrodes in a continual production process. The first machine (Grade medical´s own construction) used 2 edge electrodes with a connected voltage setting and a peristaltic pumping system of a polymer solution. The second machine had 4 electrodes with electrode settings for each one, and a pressure dosing system under argon to avoid the sublimation of the solvent from the polymer solution (experimental, subcontracted). The PCL solution for the nanofiber substrate PCLm (21 w%) was prepared by dissolving PCL in acetic acid overnight to homogenize it. The fiber production process for the tested samples was realized on the second machine. The PCL solution was loaded to a pressure dosing system with a flow rate of 594 m^3^/h. The solution was dosed on 4 edge electrodes with voltage settings of +60 kV and −60 kV, and the supported textile was moving 1 mm/min. The distance between the electrode and collector was 23 cm. The fibers were deponed in 3 layers. The polymer solution for PCL_AC was prepared from 13 w% PCL solution in acetone, by stirring overnight at a laboratory temperature. Electrospinning was realized on the first electrospinning machine with 2 edge electrodes. The parameters for the production of the nanofibers were a voltage of −60 kV and +40 kV, a rate flow of 70/38%, the distance between electrode and the collector was 24 cm, and the speed of the supported textile was 2.4 cm/min. PCL_CHE was prepared from 7 w% PCL solution on chloroform and ethanol (3:1) was stirred overnight in a closed bottle at a temperature of 40 °C. Two edge electrodes with a voltage of −65 kV and +45 kV were used for the electrospinning, the rate flow was 5/5%, the distance between the electrodes was 24 cm, and the speed of the supporting textile was 1 cm/min.

### 2.4. Solid Lipid Particles Preparation Method

SLP with an encapsulated active substance were prepared by the oil in water emulsification method. This method uses the stirring of two non-mixable liquid phases, water, and oil to prepare an emulsion. The high affinity of the active substance to the lipid phase causes its encapsulation. The preparation process is different for antioxidant SLP and antibiotic SLP.

For SLP with an antibacterial substance, the water phase consisted of 45 mL of distilled water, 0.225 mL of Tween 20, and 1.5 mL of gentamicin solution (50 mg/mL in distilled water), mixed at a temperature of 70 °C for 15 min. The oil phase was prepared from 5 g of 1-tetradecanol, mixed at a temperature of 70 °C until all the 1-tetradecanol had melted. The water phase was slowly mixed (a flow rate of 2 mL/h) into the oil phase by stirring at a speed of 600 rpm, and left at 900 rpm for 5 min, followed by a decrease in the stirring to 350 rpm and heating at room temperature. After cooling, 10 mL of the created suspension was mixed with 30 mL of distilled water, left on a shaker for 30 min set on 26 rpm, and dosed on NF PCL substrates in doses of 3.54 µL/cm^2^.

The solution for the preparation of antioxidant lipid particles was prepared from a water phase made of 45 mL of distilled water and 0.225 mL of Tween 20 and stirred for 15 min at a temperature of 70 °C. The oil phase consisted of 5 g of 1-tetradecanol, 0.3 g of tocopherol, and 5 mL of ethanol to decrease the temperature of the melting and to keep the oil phase in a liquid state during the whole process of the dosing. The prepared lipid phase was dripped into the water phase (a flow rate of 2 mL/h) by stirring at a speed of 600 rpm, and left at the stirring speed set at 900 rpm for 5 min. The heating was stopped, and the stirring speed was lowered to 350 rpm. After the cooling of the suspension at the laboratory temperature, 10 mL of the suspension was mixed with 30 mL of distilled water for 30 min. The suspension of the lipid particles was dosed on a nanofiber substrate at an amount of 3.54 µL/cm^2^.

### 2.5. Antibacterial Test

The antibacterial properties of the prepared dressings were tested on bacterial strains *S. aureus and P. aeruginosa*, which are the common bacteria in chronic wounds. Bacteria were cultivated in a lysogeny broth medium (LB) at a temperature of 37 °C for 18 h, until a concentration of 10^8^–10^9^ cells/mL was reached. One hundred µL of the LB was then added on an agar plate with circle disc samples of wound dressings of a 0.6 cm diameter. These plates were the cultivated for 20 h at 37 °C and were then evaluated as the inhibition zone of the samples to approve the antibacterial activity. For the statistical evaluation, SigmaStat 3.5 software and a one-way ANOVA test were used.

### 2.6. Tests on Biofilm

One mL of planktonic bacteria suspension with a concentration of 10^8^ KTJ/mL in Tryptic Soy Broth was placed in each well of a 12-well microtiter plate and cultivated at 37 °C for 24 h. After the cultivation, the suspension on the surface was removed and a sample of size 1 × 1 cm and 1 mL of TSB was placed on the biofilm for the continuation of the bacteria reproduction. After the cultivation at 37 °C for 24 h, the samples were removed and evaluated by inverse microscopy and a spectrometry absorbance of A_595nm_.

### 2.7. Cytotoxicity Test

The biocompatibility of the nanofiber substrates was tested with the leaching method on 3T3 fibroblasts. Nanofibers of a 6 mm diameter were put into a 220 µL DMEM medium (high glucose, D6429, Sigma-Aldrich, St. Louis, MO, USA), with 10% foetal bovine serum but without antibiotics. The samples were incubated in a CO_2_ incubator with 10% CO_2_. After 2 days, the medium from the samples was used for the experiment. Fibroblasts of 4 × 10^3^ 3T3 in a 20 µLDMEM medium with 10% fetal bovine serum were put into a 96-well plate. Subsequently, 200 µL of the extract was added into each well. Six parallel wells were used for each sample. After 24 h (D1), 48 h (D2), 72 h (D3), then 6 days, the metabolic activity was tested using an MTS assay. One hundred µL of the medium and 20 µL of the MTS solution (CellTiter 96^®^ Aqueous One Solution Cell Proliferation Assay, Promeg corp., Madison, WI, USA) was added to each well and incubated for 2 h. The absorbance was then measured at a wavelength of 490 nm and a reference wavelength of 690 nm, using a spectrophotometer Tecan Infinite M200 Pro.

### 2.8. Animal Testing

Fifty female BALB/c mice 4–5 months old were purchased from an approved licensed facility (Charles River Laboratories, Sulzfeld, Germany) and were housed in standard cages for approximately 3 weeks before entering the experiment. The investigation was approved by the Expert Committee and conformed with the Czech Animal Protection Law, the approval No. 71/2020. The animals were fed *ad libitum* with a standard diet. The skin surgery was performed under general isoflurane anesthesia. On the backs of the mice, a skin punch of the diameter 8 mm was used to prepare a full-thickness defect. Above the defect, a circle from the silicon foam of an FDA quality standard with an outer diameter of 2 cm, an inner diameter of 1 cm, and a thickness of 1 mm (kSil GP60, Gumex, Strážnice, Czech Republic) was stitched onto the skin with 6–8 sutures to avoid a spontaneous shrinkage of the skin defect. In the groups which had a bacterial infection planned, 10µL of the suspension with a bacteria mix in the phosphate-buffered saline (PBS) at a final concentration of 10^7^ CFU/mL *S. aureus* and 10^7^ CFU/mL *P. aeruginosa* was added. The defects of the groups with nanofibers (groups 2–5) were then covered with either a PCL nanofiber, PCL nanofibers with gentamicin, or nanofibers with both gentamicin and antioxidant (Table 1). In the control group 1, the uninfected wounds were not covered with nanofibers. All the wounds were then covered with 3 M Tegaderm™ and fixed with elastic adhesive bandages. After the surgery, the animals received meloxicamum 5 mg. kg^−1^s.c. and NaCl s.c. The health condition of the animals was checked every day. The nanofibers and Tegaderm were changed twice per week. On day 14, the animals were sacrificed using a cervical dislocation (after a general isoflurane anesthesia). The defects were checked, pictures were taken for a further evaluation of the wound area, and the whole defects were embedded into 4% buffered formaldehyde for histological analysis.

### 2.9. Histological Evaluation of Chronic Murine Wound Model

The whole area of the regenerated wound was collected into 4% buffered formaldehyde; after a fixation for 48 h, they were trimmed in half and moved into a 70% ethanol solution to process using an automated tissue processor (Leica ASP 6025, Leica Microsystems, Wetzlar, Germany), and embedded in paraffin blocks using a Leica EG 1150H paraffin embedding station (Leica Microsystems, Wetzlar, Germany). Sections of 2 µm and 4.5 µm were cut using a microtome (Leica RM2255, Leica Microsystems, Wetzlar, Germany) on standard glass slides. Three representative sections were stained with haematoxylin–eosin (HE) and mounted using a Leica ST5020 automated staining instrument in combination with the Leica CV5030 coverslipper. Sections on the Superfrost slides were stained with a Gram Staining Kit on the VENTANA BenchMark Special Stains System (Roche, Basel, Switzerland) for the detection of bacteria. Picrosirius Red (PSR) for the collagen quantification was stained manually.

Anti-alpha smooth muscle actin (alpha SMA) antigens were retrieved by heating the slides in an EDTA buffer with a pH value of 9. Endogenous peroxidase was neutralized with 3% H_2_O_2_. A goat anti-mouse alpha SMA monoclonal antibody (ab21027, Abcam, Cambridge, UK) with 1:200 dilutions was used as the primary antibody. The sections were incubated for 1 h at RT. After washing, they were incubated with an anti-goat secondary antibody conjugated with HRP (Zytomed Systems, Berlin, Germany). The staining of the sections was developed with a diaminobenzidine substrate kit (Dako, Agilent, Santa Clara, CA, USA) and sections were counterstained with Hematoxylinin Multistainer Leica ST5020 (Leica Biosystems, Wetzlar, Germany).

The wound dimensions, the length of impaired skin layers, the thickness of the newly regenerated skin layers, and both the thickness and length of the scar tissue, to determine the regeneration of the defect, were measured. We evaluated the morphological changes according to Simonetti [15], and the general healing according to Lazarus [16] (Table 2). For the measurement, a Zen lite 3.1 v software (Zeiss, Jena, Germany) was used. For the quantification of collagen and SMA, we used in ImageJ software, FIJI module (ImageJ 1.53f51, NIH, Bethesda, MD, USA) applying a grid of 100,000 µm^2^ and 50,000 µm^2^, respectively. The collected positive points from the whole sample were normalized to the control group. The data were statistically analyzed in a GraphPad Prism 8. We tested the normality, and subsequently performed a non-parametric Kruskal–Wallis test and Dunn’s post hoc test. The significant difference was set to *p* < 0.05.

## 3. Results

### 3.1. Structure of Wound Dressings

#### 3.1.1. Structure of Nanofiber Substrates

The nanofiber substrates for the wound dressings were prepared from three different solvents, which created three different structures (see Figure 1). The structures of the fiber substrates were analyzed by an SEM to characterize its morphology. All the samples showed a narrow fiber diameter with a combination of nanofibers and microfibers, and good mechanical properties to be used as wound covers. The material for the wound dressings needs to provide a good elasticity to cover the whole surface of the wound, and a good tensile strength to be removed from the wound. All these properties were tested during the manipulation with fiber substrates throughout the preparation process, and after the application of the suspension (see Figure 2). The structure of the substrate is important for the biological properties and can influence the cell proliferation on its surface. PCL_AC shows the beads in the structure, and there was a low polymer fiber productivity during the production process (Figure 1A). It required the application of more layers to reach the adequate thickness of the substrate. The PCL_CHE sample shows the narrow fiber without beads or other defects, but the manufacturing process brought a poor repeatability of the results (Figure 1B). The high repeatability of the medical device manufacturing process is one of the main requests. To guarantee the high quality of the manufacturing process, a high repeatability is necessary. The fibers with the smallest diameter were produced from a solution with acetic acid as the solvent. This process of the production was easy to repeat with the same final structure of the nanofiber substrate (Figure 1C).

The appropriate thickness of a nanofiber layer must be chosen based on the gas permeability on the one hand and the mechanical strength, that is essential for the manipulation, on the other hand. These properties offer a nanofiber substrate prepared by the subscribed method, with a weight per unit area of 18 g/m^2^. The resistance of the fiber substrate during the application of the lipid suspension was declared by the SEM analysis and the comparison of the structure; this is shown in Figure 2 after the application of the liquid suspension and the drying of the sample.

#### 3.1.2. The Structure of Solid Lipid Particles

The emulsification method for the solid lipid particle (SLP) preparation was tested for its repeatability by an analysis of the structure and particle size. The structure of the prepared SLP was evaluated by SEM. SLP with a spherical shape were prepared by the emulsification production process and Tween 20 was a surfactant. The choice of the surfactant has a big influence on the shape and size of the lipid particles and on the suspension stability. The average diameter of the particles was 76.6 µm and varied from 25 µm to 101 µm. The particle size was measured from the SEM images. The results from this measurement present particles on the surface of the substrate after the application. The second measurement was made using a static light scattering (SLS) analysis, which shows two peaks in the particle size distribution (see in Figure 2A), representing the particle size in the suspension (see in Figure 2B).

There were two types of particle surfaces: the smooth surface and wrinkled surface (see Figure 3). The smooth surface was produced in the particles fabricated with a higher concentration of solvent in their structure during the emulsification process and were held in a liquid phase until solidification, caused by temperature decrease. The wrinkled surface was produced during the high speed of the solidification. This process could be caused by the low local temperature of the water phase, low local concentration of the solvent during the dosing, or the high speed of the solvent dissolution into the water. The wrinkled particles offer a higher active surface for the drug release. These two types of particle surface offer a different active substance release in each particle depending on the surface, size, and amount of the encapsulated drug and the surrounding conditions.

The active substance was encapsulated in the solid part of the lipid particles and its release was based on the temperature, when the tetradecanol particle started to melt, which was at 36 °C. The inner structure of the solid lipid particle is shown in Figure 4.

### 3.2. Biological Testing of Prepared Wound Dressings

#### 3.2.1. In Vitro Experiments

We expected an increase in the inhibition zone in combination with a good biocompatibility. Microbiological testing approved the antibacterial activity of all the samples. The antibacterial activity increased with the amount of gentamicin incorporated in the structure of the SLP (Figure 5 and Figure 6) and with the amount of suspension immobilized on the substrate. The antibacterial properties were tested on the samples with different amounts of gentamicin in the SLP suspension. Three samples were prepared to show the influence of dosing an antibacterial substance during the preparation process. 

The first sample GEN_1 includes 1 mL of the gentamicin solution in the suspension, the second sample GEN_1.5 includes 1.5 mL of the gentamicin solution in the suspension, and the last sample GEN_2 includes 2 mL of the gentamicin solution in the suspension. A commercial antibacterial wound dressing StopBac Sterile Kompres with silver was used as a positive control (C+), and a PCL substrate without a suspension was used as a negative control (C−). The samples are more effective on the *S. aureus*, where the inhibition zone of the sample GEN_2 is more than 5 mm (see Figure 6).

An obstacle during the healing of the infected wounds is biofilm that can be described as a microbial colony, encased in a polysaccharide matrix on the surface of the wound. Chronic wounds particularly appear to be more susceptible to the biofilm formation. James’ study shows that 60% of the chronic wounds have a biofilm, in comparison with 6% of the acute wounds [17]. Therefore, antibacterial testing was extended for the efficiency on the bacterial biofilms of *S. aureus* and *P. aeruginosa*.

All the samples were effective in the eradication of biofilm formed by the bacteria strains *S. aureus* and *P. aeruginosa*. In the control wells STAU_C and PSAE_C with PCL substrate without the SLP suspension Gen, there was a large bacterial growth in comparison with the wells with the functionalized samples STAU_GEN and PSAE_GEN, where we can see a large eradication of the biofilm. In these wells, we can see only the planktonic cells or small clumps of bacteria (see Figure 7), but there is no continual layer of non-dissolved biofilm under or around the samples in the whole well.

The spectrometry analysis of the biofilms after the application of MA_Gen_1.5 samples shows that the absorbance decreased by 85% at the STAU and by 86% at the PSAE after the application of the functionalized substrate (see Figure 8).

The in vitro cytotoxicity testing of the nanofiber substrates showed a significantly lower metabolic activity of the 3T3 fibroblasts on both the PCL_AC and PCL_AC_GEN, which were prepared using acetone as a solvent (Figure 9). PCL is a biocompatible material. The cytotoxicity may have been caused by the remnants of acetone in the nanofiber substrate, although all the substrates were left free to allow for the evaporation for at least 7 days.

Contrarily, the nanofiber substrates prepared from both chloroform and acetic acid were not cytotoxic and supported the cell growth and metabolic activity. Subsequently, fibres made from acetic acid solvent were used for the in vivo experiments.

#### 3.2.2. Active Substances Release 

The release of gentamicin was tested on the samples with a nanofiber substrate with SLP with gentamicin GEN_1.5. The samples of size 4 cm^2^ were leached at a temperature of 37 °C, in 5 mL of physiological fluid (0.9% NaCl solution) for 24 h. Physiological fluid simulates the environment of the wound, and by using the LC-MS analysis, the concentration of the gentamicin in the solution was measured. The final concentration of gentamicin after 24 h of leaching was evaluated. After 24 h in a simulated environment, 91% of gentamicin from the lipid structure was released, which created a concentration of 903.9 ng/mL

#### 3.2.3. In Vivo Experiments

In all the defects, a silicon ring was used to slow down the shrinkage of the defects and, to a limited extent, simulate the non-healing wound conditions. *S. aureus* and *P. aeruginosa* were used to emulate the bacterial infection. In the control group, the healing of the defect was fast as it was almost healed on day 14 (Figure 10). In some animals, the defects were partially covered with fur. In the PCL + Gen + VitE group, which did not contain bacteria, the defect continuously decreased and after 14 days, the defects were almost healed, which was similar to the control group with the presence of a small scab and fur on the defect (Figure 10). The defects in the PCL + Bac group healed more slowly and the defects were bigger than that of the control group. Some of the defects were filled with the granulation tissue and scab. The PCL + Gen + Bac group showed bigger differences, mainly on day 14. In some animals, the defects were almost healed, however in other animals, there was a scab the size of the original defect or bigger, and the defects contained pus. The defects in the group PCL + Gen + Bac + VitE healed slowly, the defect size decreased, and they were filled with the granulation tissue and/or scab. (Figure 10). The area of the defects was measured from the pictures taken on days 7, 10, and 14 (Figure 11).

#### 3.2.4. Histology Evaluation

Six animals from the different groups with applied bacteria did not survive until the last experimental day, and one sample was destroyed during the extraction process (Kaplan–Meier survival analysis in Appendix A). 

The measurement of the newly formed skin suggests that the bacterial infection prolongs the constriction and regeneration of the wound, despite no significant differences among the groups. We can see a non-significant difference in the group treated with enriched fibers where the shortest non-complete skin appeared, signaling a promising effect of PCL + Gen + VitE combination. The presence of bacteria in the wound, significantly increased the thickness of the scar as well as the higher hypertrophy of the newly formed skin layers (Figure 12). In the groups with an induced bacterial infection, epidermis grew multifold in the thickness. We observed similar results in the dermis and hypodermis, where hypodermis in the group treated with PCL + Gen + VitE + Bac formed almost twice the thickness compared with the control groups without an induced bacterial infection (Figure 12, Figure 13, Figure 14 and Figure 15). All the samples showed at least an acceptably healed wound (Healing Status from Figure 13).

Figure 13 explains the multiple aspects of the histological evaluation that characterizes the healing of the wound. The newly formed tissue reached the adequate thickness and the keratinocytes migrated into the outer parts of the wound, creating an epidermis as seen in the evaluation of the re-epithelialization. The stratum corneum often completely formed in the control groups without the applied bacterial mix (Figure 14 and Figure 15). Significant differences occurred between the control group and PCL + Gen + VitE + Bac, and also between PCL + Gen + VitE + Bac and the uninfected PCL + Gen + VitE. The dermis and hypodermis stayed mostly in the tissue granulation phase; however, the determination of dermis was problematic due to the incomplete layer formation. A significantly higher dermis organization was found in the control group compared to both the PCL + Gen + VitE + Bac and PCL + Bac groups, while in the evaluation of the tissue granulation, significant differences were similar to the re-epithelialization, again between the PCL + Gen + VitE + Bac and the control group, and the PCL + Gen + VitE + Bac and the uninfected PCL + Gen + VitE group. The inflammation of the tissue was, as expected, more prominent in the infected groups even despite the added antibiotics, which was the same as the presence of bacteria in the newly formed tissue; the results of the evaluation of the inflammation and the presence of bacteria followed the same pattern as the results of the re-epithelialization and tissue granulation, with significant differences between the PCL + Gen + VitE + Bac and control group, and the PCL + Gen + VitE + Bac and uninfected PCL + Gen + VitE group, also presented in Figure 13. A higher amount of collagen fibers and a lower number of cells appeared in the control groups without an induced bacterial infection, yet they lacked significant differences (Figure 13). The representative images illustrate newly formed tissue in Figure 14 and Figure 15. The neovascularization marker SMA did not show any significant differences among the groups, but it often varied within them (Figure 13, Figure 14 and Figure 15). The adipose layer of the hypodermis rarely fully formed, even though single adipose cells appeared, yet papillae emerged in the groups missing a bacterial infection. 

## 4. Discussion

The regeneration of a chronic wound is a complex process that requires time and an appropriate environment. A clean wound with granulating tissue and lacking an infection, together with a moist wound dressing, seems to be the best recipe for a successful healing. The TIME method promises to be an effective tool to deliver such conditions. Devitalized tissue needs to be removed, either local or systemic infections eliminated, a proper moisture balance set using the various dressing options, and, if the desired result of healthy healing still does not appear, an edge advancement such as tissue substitutes could be applied to help with the cell migration and proper layer formation [18,19].

Our results brought a positive outcome, showing that the regeneration of a clean wound without a bacterial contamination was not negatively affected by the application of either hydrophobic PCL or PCL enriched with gentamicin and vitamin E. The results were very similar in all the tests, which demonstrates the biocompatibility of the selected material in skin wound healing. PCL nanofibers are biocompatible and support the proliferation of fibroblasts, keratinocytes, and melanocytes, but cell growth is influenced by the composition of a medium, FBS, or the platelet lysate content [20]. Skin cell growth is also influenced by the nano/microtopography of the fibers. The keratinocytes grew better on the nanostructured PCL, while the fibroblast grew well on both the 2D nanofibers and 3D microfibers [21].

In our experiment, the addition of antibiotic and antioxidant substances did not cause a significant change in the size of the wound, but it was able to provide a better formation of all three skin layers, and even the thinner layers when compared with the control group without an induced infection. A faster closing of the wound and the formation of skin layers leads to a quicker structural and physiological gain of the function, an important aspect of good healing.

The granulation of the tissue and dermis organization seem to prosper with added gentamicin and vitamin E enrichment, compared with the PCL fibers only. Gentamicin did not completely eradicate the effect of the bacterial inflammation present in the wound, yet the histological score of the reepithelization, granulation of the tissue, and dermis organization rose, while the marker of the tissue inflammation suggested its positive effect. The effect of the gentamicin and VitE released from the lipid layer on the fibers seems inadequate in the case of the bacterial infection of the wound. However, the application of the bacterial mix of *S. aureus* a *P. aeruginosa* caused an inflammation serious enough to lead to a preterm death of the experimental animals, while those with added antibiotics persevered the experimental settings.

Olekson and the team studied the effect of a direct topical application of an ultrahigh dosage of gentamicin as a treatment of the skin grafts in a porcine model. They proved that gentamicin in an ultrahigh dosage can obstruct the neovascularization in the wound, demonstrated by the downregulated markers of angiogenesis, and upregulated the markers of the proinflammatory response a week post-grafting. However, they observed neither specific differences in the time of healing, nor a contraction of the wound during the course of the healing [22]. Similar results come from Junker et al., where they proved in vitro the negative effect of a high dosage of gentamicin (1000 µm/mL) on the proliferation of cells, human keratinocytes specifically. Their team also focused on the topical application of gentamicin in various concentrations in a porcine model of the wound, and on further analyses of the blood, the wound fluid for the inflammatory and antibacterial response [23]. In comparison with these studies, we did not overshoot the gentamicin concentration, as smooth muscle actin, a marker of vascularization, was increased in the samples with gentamicin and we observed the migration of the keratinocytes, and a more defined formation of the epidermis. Our results also suggest that the addition of gentamicin to the SLP formulation can slow the release of gentamicin and stabilize its concentration, even in the presence of a bacterial infection. In the same comparison, we were able to see the interindividual variability of the healing, even in the presence of a bacterial infection with the exception of the SMA. The marker of vascularization varied more in the presence of gentamicin during a raging bacterial infection, but its presence showed its positive effect the on the new formation of vessels, despite the bacterial inflammation. All these results support the findings of Abdul Khodir’s group, claiming that gentamicin could also promote cell viability in vitro when blended directly with PCL and collagen, an important building block of the extracellular matrix, to form a fiber [24]. Kimna even confirmed the positive effect of gentamicin released from plant-based nanofibers in vitro [25].

There is a notable difference between the studies supporting our findings and our experimental settings. In the aforementioned studies, gentamicin was either directly added or mixed with the polymer prior to the fabrication, while in our case, the antibiotic-antioxidant mix was deponed on the PCL fibers in a lipid emulsion. The antimicrobial effect of the used formulations was proved in vitro. In animals, it seems that the bioavailability of the drug should be higher and its effect more prevalent, since PCL does not need to dissolve to release gentamicin from the fibers. Moreover, the release of gentamicin and VitE was supported by the increased temperature at 37 °C.

According to our hypothesis, an additional enrichment of the fibers with VitE, for which the most prevalent form is α-tocopherol, should have promoted the healing of a chronic wound. Unfortunately, we could not confirm this. Our results could be explained in multiple ways. One could be by the technical limitation of the experimental set up. 

Since VitE is soluble in fats, we used a lipid emulsion for its delivery. We chose myristyl alcohol, otherwise known as 1-tetradecanol, a fatty alcohol widely used in cosmetics that, when applied topically, proved to be effective in the fight against bacteria which induced periodontitis in rabbits. In a 100 mg/mL concentration, it exhibited anti-inflammatory properties, arrested the inflammation progress, and even promoted the restoration of the periodontal tissue [26,27]. Despite previous promising facts, a further addition of VitE in combination with PCL, myristyl alcohol, and gentamicin did not significantly decrease the inflammation caused by the bacterial infection, nor increase the histological score towards the healing, when compared with the results of the groups without VitE nor gentamicin. Furthermore, we observed no negative changes comparing the results of the control groups without an induced infection and nanofiber, and an infection free control with Gen + VitE-coated PCL. The group with PCL + Gen + VitE showed an even slightly more formed dermis.

Despite the general impression in the 1980s and 1990s of the use of natural herbs and supplements for the treatment of skin defects and many other problems, the beneficial effects of VitE on skin wound healing were not proved [28]. Harri Hemilä and his team studied the effect of vitamin E supplementation in older male smokers in the ATBC study from southwestern Finland. During the three years of 1985–1988, 29,133 men aged 50 to 69 years, who smoked at least five cigarettes per day, joined the study. They were randomly assigned to four groups and took either α-tocopherol as dl-α-tocopheryl acetate (50 mg/day), beta-carotene as all-trans-beta-carotene (20 mg/day), both supplements, or placebo capsules for 5–8 years (median 6.1 years). A post-intervention follow-up has continued through the Finnish Cancer Registry and other national registries, and epidemiological analyses continue to be conducted. Depending on the age, lifestyle, number of cigarettes smoked per day, and other aspects of the participants, they observed that pneumonia occurrence decreased up to 72% in 50–69 years old [29], mortality lowered in a group of men aged 66–69 up to 41%, while it increased up to 19% in the case of 50–62-year-old smokers [30]. Out of multiple analyses, the author strongly argues that the effect of vitamin E immensely varies depending on one’s age, health status, and other aspects, and that defining specific subgroups where vitamin E supplementations benefits its consumer is crucial [31].

In 2016, Tanayidin et al. reviewed the topical application of VitE on skin scars and wound healing with unflattering results for VitE. They concluded that monotherapy by topical VitE application has not yet gathered sufficient evidence of its significant beneficial effect on scar appearance to justify its widespread use [32]. However, 2018 brought multiple studies which focused on an α-tocopherol encapsulation in the lipid nanomaterials, confirming the promotion of skin wound healing. Caddeo and their team were able to see the antioxidant effect of the transferosomes, soy phosphatidylcholine-Tween based nanovesicles, loaded with a-tocopherol acetate, against hydrogen peroxide, and the permeability of such particles through the porcine skin ex vivo. These transferosomes even promoted cell proliferation and the migration of epidermal keratinocytes (HaCaT) and 3T3 dermal fibroblasts, which resulted in an acceleration of the skin wound closure [33]. Bonferroni encapsulated α-tocopherol into chitosan oleate, amphiphilic polymer lipid salt, and observed a statistical difference in the keratinocyte proliferation in α-tocopherol-loaded chitosan oleate compared with the chitosan oleate group at 24 h. During a 7-day incubation, the cells on chitosan oleate reached the values of the α-tocopherol-loaded group though [34], suggesting either a lower involvement of α-tocopherol in the later keratinocytes proliferation, a lower bioavailability of the active molecule, or a general lack of concentration of the molecule on day 7. These papers suggest that a proper choice of lipid is crucial for α-tocopherol to succeed. Horikoshi even found a signaling pathway of keratinocytes polarization through the α-tocopherol regulation of Par3 and aPKC localization, and a complex formation during the wound healing [35]. Na et al. created self-assembled ferrocene nanocapsules loaded with α-tocopherol. The nanocapsules demonstrated ROS-scavenging properties on 3T3 fibroblasts, and even healing properties, which were tested on a scratch assay [36]. Horikoshi’s results would confirm our results of a more defined derma since keratinocytes move topically, yet we cannot definitely confirm the rest of the studies. Vitamin E is increasingly recognized as a modulator of the immune system and inflammation, as Lewis et al. widely discussed in their review [37]. In 1980, Prasad proved that 300 mg/day in the form of dl-α-tocopherol acetate administered to 18 young males, 13–30 years old, for 3 weeks, stabilized the leukocytes cell membrane up to the point where the bactericidal activity decreased [38,39,40]. According to multiple authors, VitE often accompanies allergic reactions [41,42,43,44].

Naguib and Valvano reported that both water soluble and liposoluble forms of vitamin E inhibited the binding of the bacterial lipocalin protein BcnA, produced by Gram-negative bacteria *Burkholderia Cenocepacia*, to antibiotics. Subsequently, the increased MIC of norfloxacin and ceftazidine was shown in vitro [45]. The inhibitory effect of α-tocopheryl acetate on the formation of the biofilm in vitro, using different bacterial strains, e.g., *S. aureus*, *S. epidermidis*, *P. mirabilis*, and *P. putida,* was proved by Vergalito [46]. The authors also showed a decreased biofilm formation by *S. aureus* and *S. epidermidis* on the urinary catheters modified with VitE. Amevor et al. showed that breeding hens supplemented with quercetin, and/or vitamin E, significantly improved the egg quality, significantly increased the serum immunoglobulins IgA, IgM, and IgG and cytokines (IFN-γ and IL-2) concentrations, and increased the expression of the splenic immune-related genes IL-2 and INF-γ, compared to the control [47].

Saturated fatty alcohols, such as myristyl alcohol (1-tetradecanol), were reported to enhance the penetration of melatonin through the skin [48]. The highest penetration of melatonin through the rat skin was observed with decanol, and with the increasing carbon chain length slightly decreased [49]. However, the positive effect on the penetration of lipophilic drugs is accompanied with the irritation of the skin. The transepithelial water loss (TEWL) of tetradecanol was the highest (48.4 g.m^−2^) among other alcohols after 48 h. The erythema score of tetradecanol (1.3.) increased for decanol (1.7) to dodecanol (2.0) and decreased in tridecanol (score 1.0) and dodecanol (score 1.3) [49]. In tridecanol and tetradecanol, a delayed irritation was observed due to their lipophilic character, a fast diffusion into the stratum corneum, but a slower diffusion into the aqueous epidermis [49].

Another explanation, much simpler, presents itself to explain the poorer results of VitE in the presence of a bacterial infection. For the appropriate function of innate immunity towards an infection, free oxidative radicals are a necessary weapon. As Wink et al. [50] widely discussed and clarified, the role of redox molecules such as ROS and NO are crucial in mammal immunity and even in the regulation of tissue repair. The combination of reactive oxidative and nitrogen species creates an effective bactericidal tool. In the case of an infected wound, such as our model with an applied mix of *S. aureus* and *P. aeruginosa*, an antioxidant effect of VitE as a scavenger of free radicals might have delayed a proper immune reaction and worsened the healing process. In addition, Niki argues that α-tocopherol mainly scavenges peroxyl radicals (RO_2_•), suggesting its role in the infected healing wound [51]. However, the α-tocopherol specificity of scavenging RO_2_ radicals might not be the primary reason for our poorer results.

Wink et al. mentions that NO, a redox immunomodulator involved in the inflammation, tissue regeneration, cell proliferation, restructuring of the extracellular matrix, and even revascularization, is a lipophilic molecule [50]. On reflection of the generally poorer results from the histological evaluation analyses, an application of PCL fibers, a hydrophobic structure on one side, and a lipid layer of SLP loaded with active molecules on the other, a model of a chronic wound with an induced bacterial infection might not offer the best environment to eradicate the infection. Yet, a combination of PCL, gentamicin, and VitE in a clean, non-infected wound did not worsen the healing process. These results agree with our study of an acute skin defect treatment, where PCL nanofibers showed worse results than hydrophilic polyvinyl alcohol nanofibers [52]. A similar treatment in a porcine model of the chronic wound is currently being performed to confirm these results.

In addition to the results discussed above, which are fundamentally dependent on the specific selected active substances, the experiments contributed to the verification of the feasibility of the concept of the preparation of wound dressings in outpatient conditions using a suitable functionalization device. Individual samples showed consistent monitored parameters (e.g., biocompatibility) and thus indicate repeatability, which is essential for the clinical use of any medical device. The concept of personalized care is being scientifically developed more intensively, and the first outputs are already appearing on the market. The company Nanomedic Technologies Ltd. launched the SpinCare device on the market, which is intended to be used to produce personalized nanofiber wound covers with the option of containing an active component in the nanofiber mass [53]. Although this system shows certain advantages from the perspective of ambulatory application, the possibility of personalizing individual covers is limited (the possibility of using only pre-prepared cartridges containing a spinning solution with an active substance) and the created layer can only guarantee some essential parameters, e.g., barrier function to a very limited extent. In contrast, a system using a mass-produced nanofibrous carrier with guaranteed parameters, which is functionalized on a device allowing the setting of a number of parameters (in particular, the content of the active substance in the particles and the total volume of deposited particles), provides a fundamentally greater added value in this respect.

## 5. Conclusions

Nanofibrous substrates enriched by gentamicin and tocopherol acetate encapsulated in solid lipid particles were prepared and assessed with the aim of proving their potential for supporting the effective healing of a chronic wound. We successfully simulated the chronic wound in a murine model by the combined application of a bacterial mix of *S. aureus* and *P. aeruginosa* and silicone O-rings as holders to limit the physiological closing of the regenerating rodent skin. From our results, the eradication of a bacterial infection is the most important step for a proper healing and wound closure. An early application of the antioxidant molecule VitE, bound to the PCL nanofibrous substrate by a lipid linker into the infected wound, did not accelerate the healing process nor shorten the inflammation to confirm our hypothesis. We observed a lower area of uninfected defects compared to the infected defects. Fourteen days after surgery, the PCL + Gen + VitE + Bac groups showed a lower inflammation score, granulation, and reepithelization, compared to both the uninfected groups. Moreover, we observed a lower dermis organization compared to the control, a higher scar thickness compared to the other groups, and a higher thickness of the hypodermis and bacteria score compared to both the uninfected groups. We conclude that the tocopherol acetate should not be applied before the eradication of a bacterial infection in the treatment of chronic wounds. We suggest a further examination of the proper combination of PCL, gentamicin, and tocopherol acetate, or other forms of the vitamin E group, regarding the timing dependent on the wound status for the treatment of the chronic wound, to attain the best results. However, this study demonstrated the selected basic parameters of the safety (biocompatibility) of the designed wound cover as a medical device, and the indicated feasibility of the concept of the preparation of wound dressings in outpatient conditions using a suitable functionalization device.

## Figures and Tables

**Figure 1 nanomaterials-12-03824-f001:**
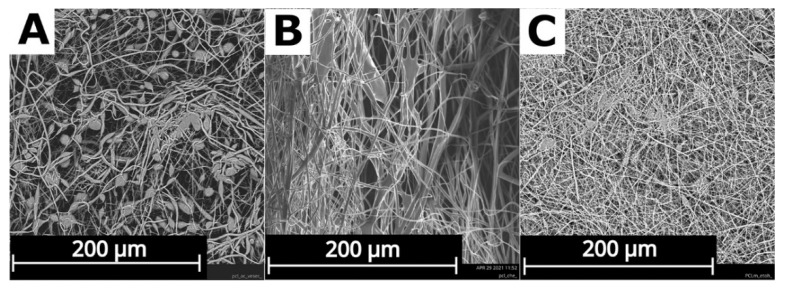
Structures of polycaprolactone substrates from different solvents (**A**) acetone (PCL_AC), (**B**) chloroform + ethanol (PCL_CHE), and (**C**) acetic acid (PCL_AA). Scanning electron microscope (SEM) analysis.

**Figure 2 nanomaterials-12-03824-f002:**
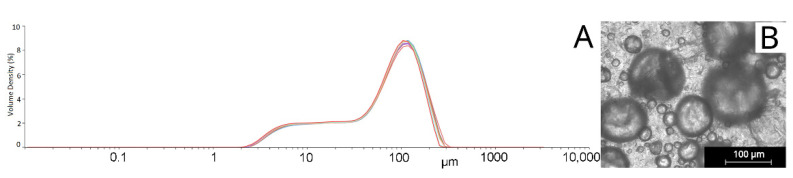
Static light scattering analysis of solid lipid particles (SLP) in suspension GEN_1.5 (**A**) SLP size distribution, and (**B**) SLP before immobilization.

**Figure 3 nanomaterials-12-03824-f003:**
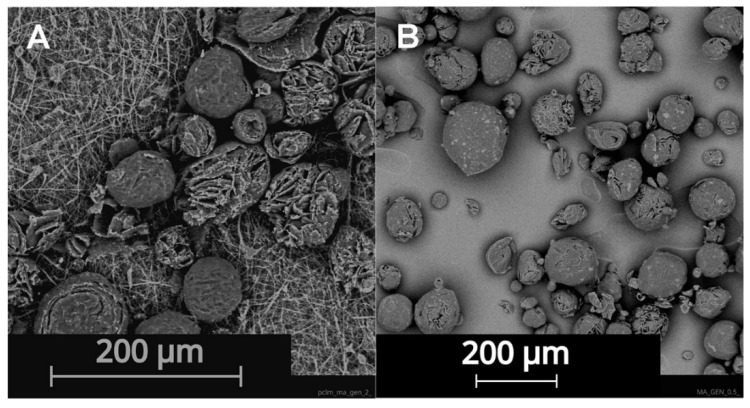
Structure of SLP placed (**A**) on nanofiber substrate PCL_AA (**B**) on tissue culture plate. SEM analysis.

**Figure 4 nanomaterials-12-03824-f004:**
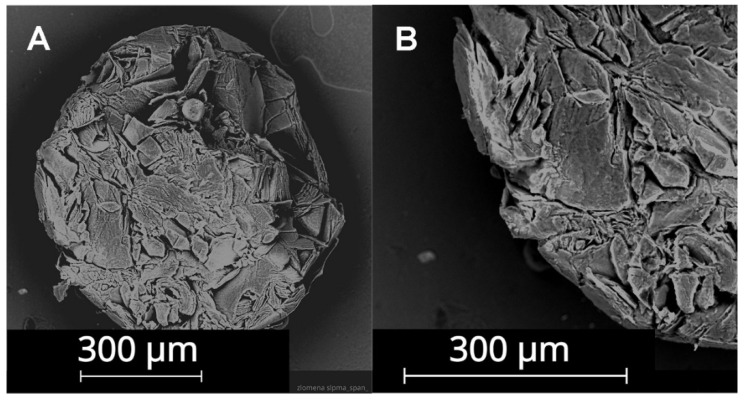
Structure of tetradecanol. Magnification at (**A**) 300×, (**B**) 550×. SEM analysis.

**Figure 5 nanomaterials-12-03824-f005:**
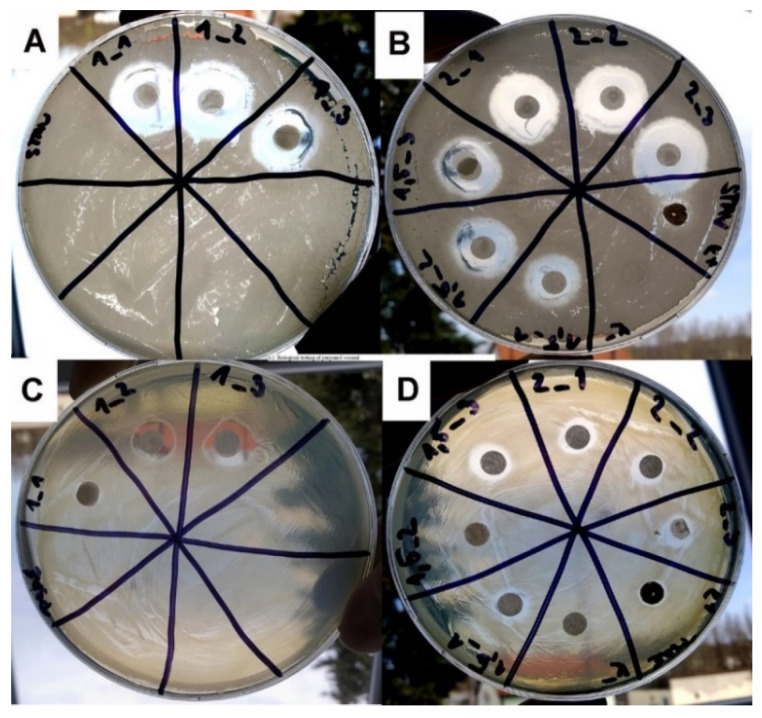
Agar plates for antibacterial tests with bacterial strains (**A**,**B**) *S. aureus* (SA) and (**C**,**D**) *P. aeruginosa* (PA). Samples GEN_1 (in 3 repetitions 1_1, 1_2, 1_3), GEN_1.5 (in 3 repetitions 1.5_1, 1.5_2, 1.5_3), and GEN_2 (in 3 repetitions 2_1 2_2, 2_3). Inhibition zones around samples were measured to compare their antibacterial activity.

**Figure 6 nanomaterials-12-03824-f006:**
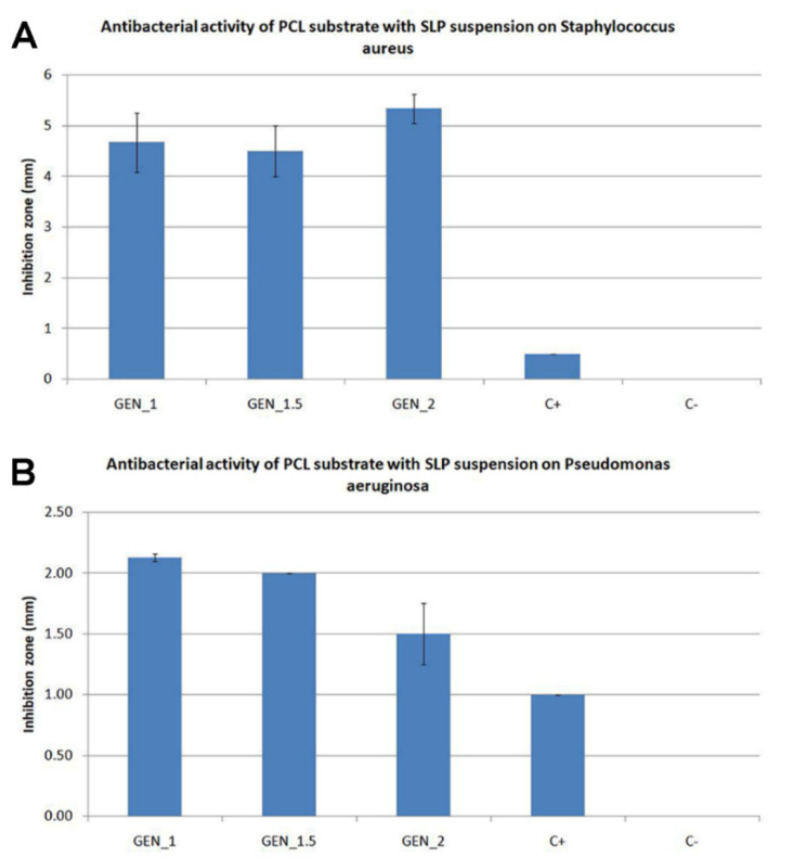
Influence of doses of antibacterial SLP with different volumes of gentamicin (GEN) solution substance on each bacterial strain (**A**) *S. aureus* (SA) and (**B**) *P. aeruginosa* (PA) after 20 h of incubation on agar plate at temperature 37 °C, C+ = control with silver, and C− = PCL substrate without lipid suspension. The statistical analysis shows there is not statistically significant difference for STAU (*p* = 0.152) or for PSAE (*p* = 0.164), (STAU—*S. aureus*, PSAE—*P. aeruginosa*).

**Figure 7 nanomaterials-12-03824-f007:**
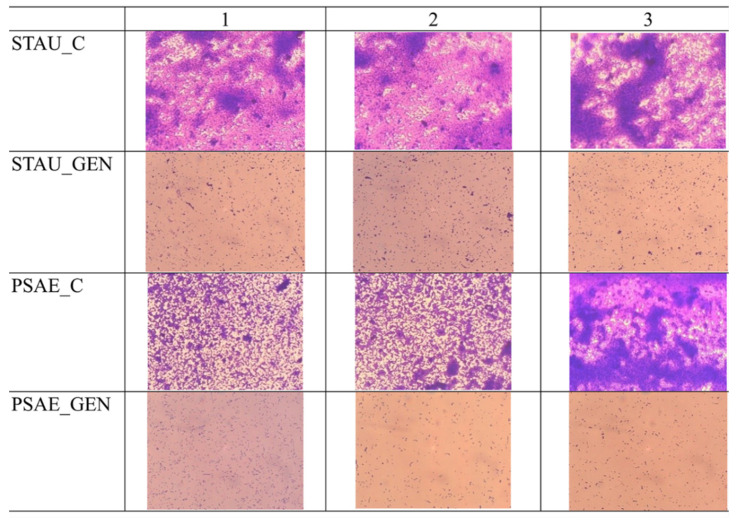
Eradication of biofilm of *P. aeruginosa* (PSAE_C) and *S. aureus* (STAU_C) by PCL substrate with SLP from myristyl alcohol was not visible but PCL substrate with myristyl alcohol and gentamicin (MA_GEN_1,5 group) (PSAE_GEN, STAU_GEN) showed sufficient bacteria eradication (scanned by inverse light microscope analysis), the samples were tested in triplicates and the results are shown in 3 columns.

**Figure 8 nanomaterials-12-03824-f008:**
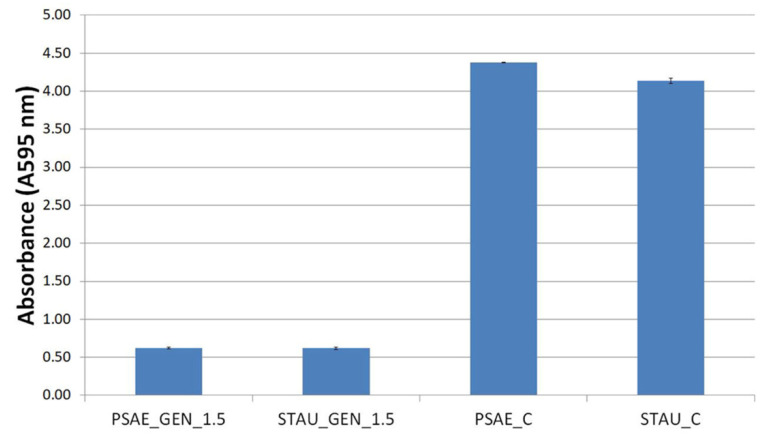
The eradication of biofilms by MA_GEN_1.5 measured by spectrometry. These results showed that the PCL nanofibers substrates with SLP_GEN_1.5 not only prevents biofilm growth but eradicates biofilm structure. *P. aeruginosa* (PSAE) and *S. aureus* (STAU).

**Figure 9 nanomaterials-12-03824-f009:**
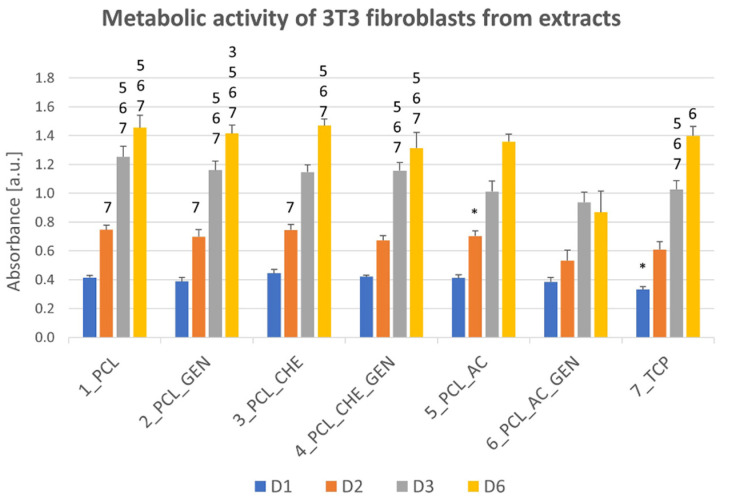
Biocompatibility of extracts of pure nanofiber substrates prepared with different solvents (PCL, PCL_CHE, PCL_AC) and extracts of nanofiber substrates functionalized with solid lipid particles with gentamicin (GEN), tested with murine 3T3 fibroblasts on days 1, 2, 3, and 6. Statistical significance between the groups was set at *p* ˂ 0.05 and is described in the above columns. * shows the statistical difference compared to all other samples, a number represents the number of the sample.

**Figure 10 nanomaterials-12-03824-f010:**
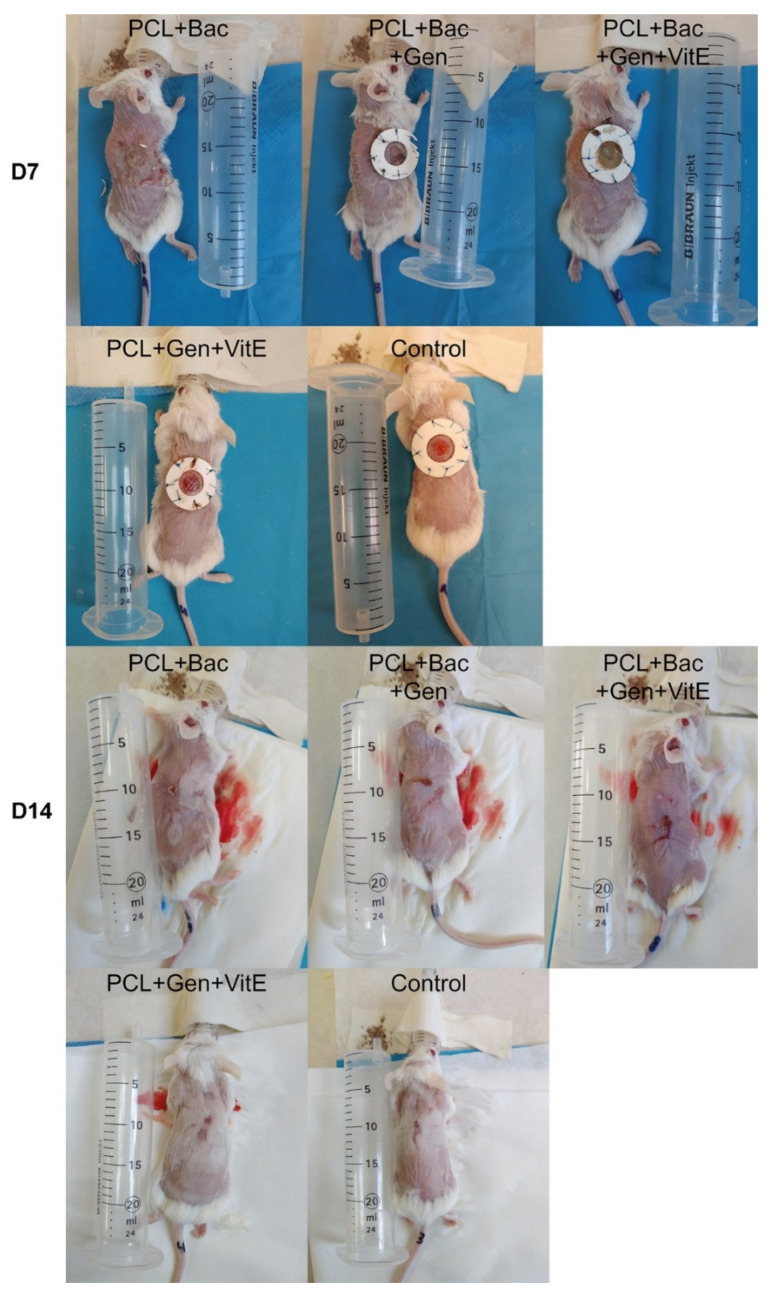
Skin defects of the mice treated with PCL nanofibers (PCL), gentamicin (Gen), α-tocopherol acetate (VitE), and infected by *S. aureus* and *P. aeruginosa* (Bac) or untreated defects (control) on day 7 and 14 after surgery.

**Figure 11 nanomaterials-12-03824-f011:**
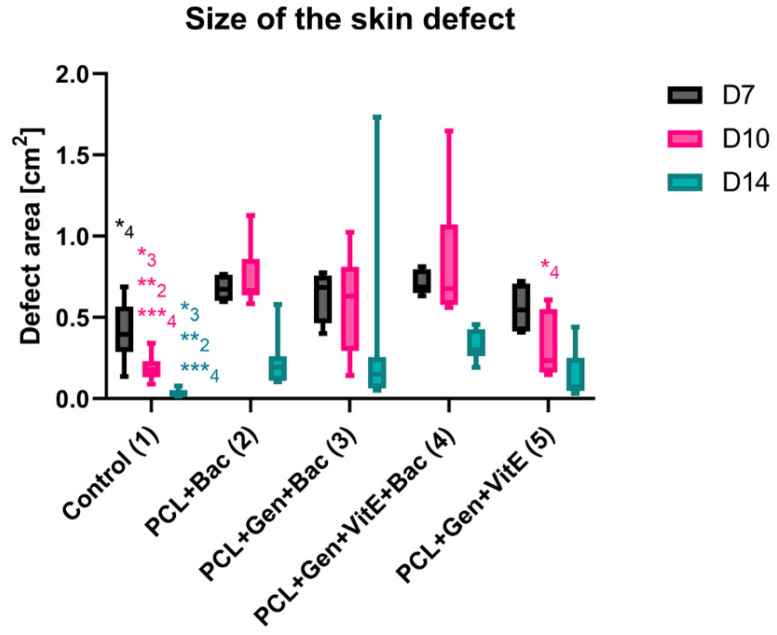
The size of the skin defects of mice treated with PCL nanofibers (PCL), gentamicin (Gen), α-tocopherol acetate (VitE), and infected by *S. aureus* and *P. aeruginosa* (Bac) or untreated defects (control) on days 7, 10, and 14 after surgery. The statistical significance was marked as a number representing the group. Significant difference * *p* < 0.05, ** *p* < 0.01, *** *p* < 0.001.

**Figure 12 nanomaterials-12-03824-f012:**
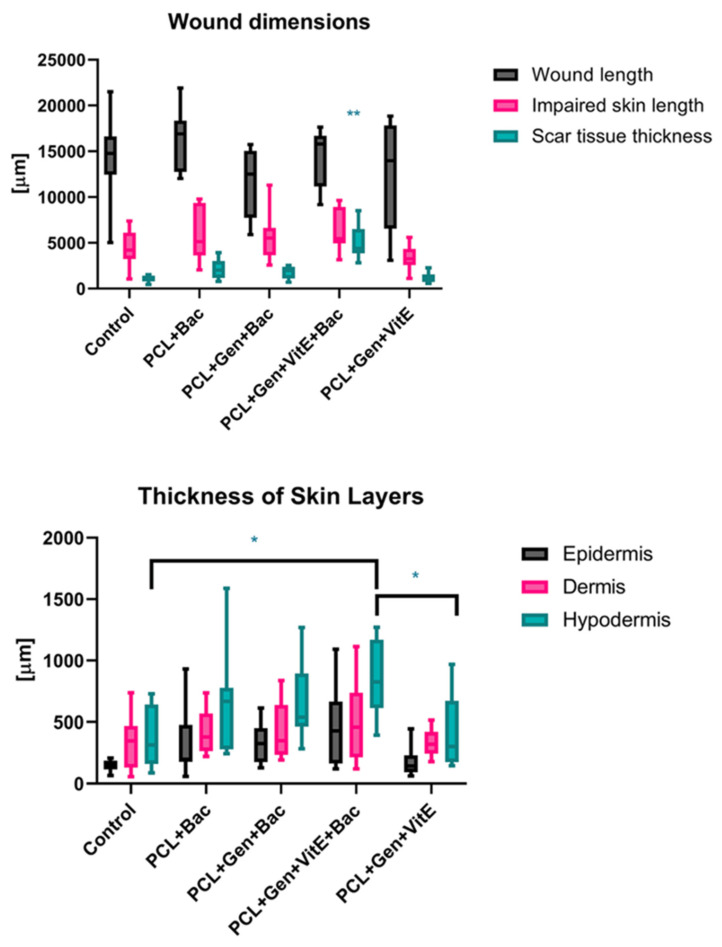
Histological evaluation of newly formed tissue according to Table 2. Defects treated with PCL nanofibers (PCL), gentamicin (Gen), α-tocopherol acetate (VitE), and infected by *S. aureus* and *P. aeruginosa* (Bac) or untreated defects (control). Significant difference * *p* < 0.05, ** *p* < 0.01.

**Figure 13 nanomaterials-12-03824-f013:**
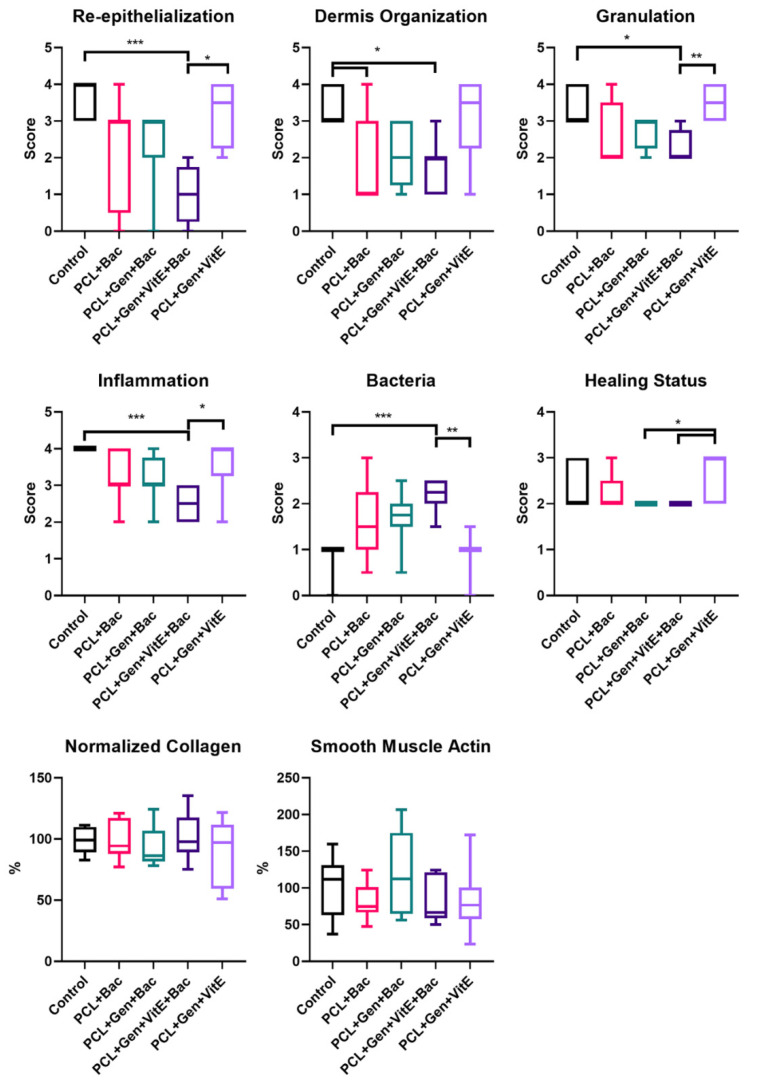
Histologic evaluation according to Table 2. Defects treated with PCL nanofibers (PCL), gentamicin (Gen), α-tocopherol acetate (VitE), and infected by *S. aureus* and *P. aeruginosa* (Bac) or untreated defects (control). Significant difference * *p* < 0.05, ** *p* < 0.01, *** *p* < 0.001.

**Figure 14 nanomaterials-12-03824-f014:**
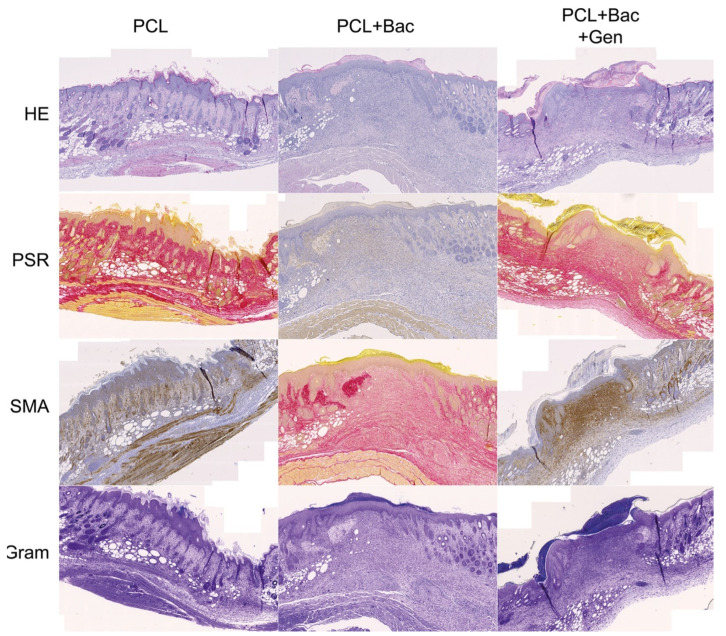
Newly formed skin, tissue 5 × 3.33 cm, scale bar 500 µm. PCL—polycaprolactone, Bac—induced bacterial infection by *S. aureus* and *P. aeruginosa*, Gen—gentamycin, HE—haematoxylin–eosin, PSR—picrosirius red, and SMA—smooth muscle actin alpha.

**Figure 15 nanomaterials-12-03824-f015:**
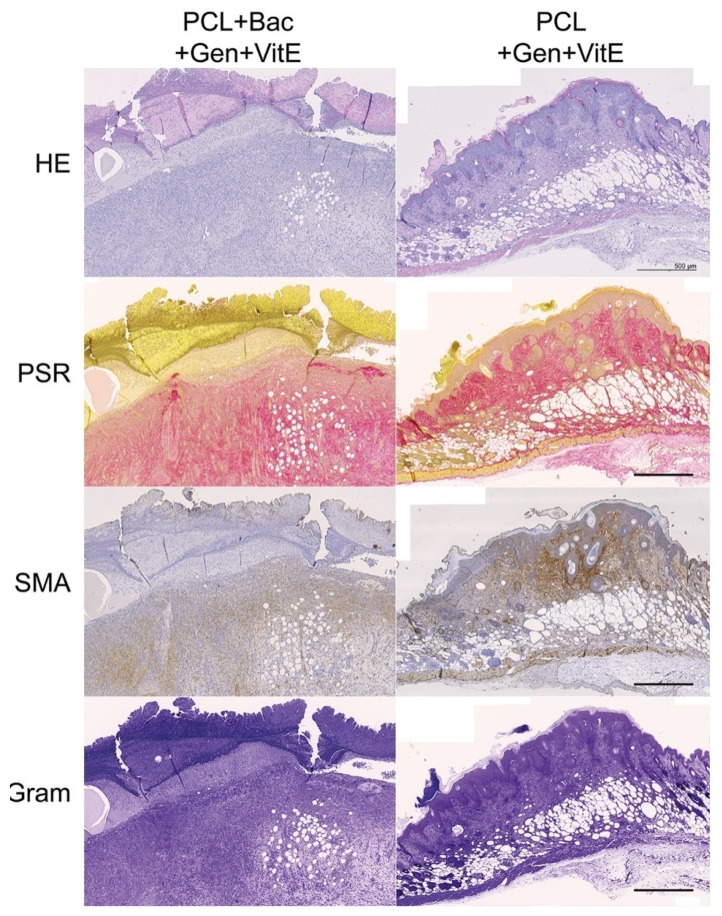
Newly formed skin, continuation of Figure 14, tissue 5 × 3.33 cm, scale bar 500 µm. PCL—polycaprolactone, Bac—induced bacterial infection by *S. aureus* and *P. aeruginosa*, Gen—gentamycin, HE—haematoxylin–eosin, PSR—picrosirius red, and SMA—smooth muscle actin alpha.

**Table 1 nanomaterials-12-03824-t001:** Experimental groups of mice in in vivo experiment.

Group No.	Name of Group	Description
1	Control group	Wound without bacteria and nanofiber substrate
2	PCL + Bac	Wound with bacteria strains (*S. aureus, P. aeruginosa*) and covered by nanofiber substrate from polycaprolactone (PCL)
3	PCL + Gen + Bac	Wound with bacteria strains covered by nanofiber substrate with gentamicin (Gen)
4	PCL + Gen + VitE + Bac	Wound with bacteria strains covered by nanofiber substrate with Gen and α-tocopherol acetate (VitE)
5	PCL + Gen + VitE	Wound without bacteria strains covered by nanofiber substrate with Gen and VitE

**Table 2 nanomaterials-12-03824-t002:** Histologic evaluation of morphological changes and wound healing. (Adapted by Simonetti et al. [15] and Lazarus et al. [16]).

Score	Reepithelialization	Granulation Tissue	Dermis Organization	Inflammation	Wound Healing	Bacteria
0	Trace and modest keratinocyte migration	Trace	Absent	Absent	None	None
1	Evident keratinocyte migration	Hypocellular and no vessels	Trace	Modest	Minimal	+
2	Differentiation	Many cells and few vessels	Modest	Good	Acceptable	++
3	Hypertrophic and partial stratum corneum	Many fibroblasts, some fibers, and some vessels	Good	Evident	Ideal	+++
4	Complete and normal	More fibers and few cells	Evident	Absent	-	-

## Data Availability

Not applicable.

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
