# Peer review of "The Preparation and Biological Testing of Novel Wound Dressings with an Encapsulated Antibacterial and Antioxidant Substance"

_nanomaterials, 2022, doi:10.3390/nano12213824_

Round 1
Reviewer 1 Report
Remarks
1. Line 79 - the acronym ROS has already been introduced previously in the line 68, so there is no need to redefine it.
2. Line 96 - PCL acronym requires some explanation.
3. Line 122 - "Penta chemicals" the second part of the name has a lowercase letter, while on line 123 the uppercase letters have been used. Please standardize the spelling of the names of companies producing chemical reagents throughout the work.
4. Line 131 - why is “Scanning electron microscopy” capitalized?
5. Line 152 - notation of the unit 594 m3/h incorrect - the number 3 should be in the exponent.
6. Line 173 - the SI system of units applies, so the temperature should be written in Kelvin. Please standardize the entry throughout the work.
7. Line 207 - it should be "6 mm".
8. Line 213 - it should be “48h (D2), 72 h”.
9. Line 372 – it should be “Figure 7”
10. Figure 7 - What do the columns 1, 2, 3 mean in figure 7?
11. Line 575 - it should be “[21]. HarriHemilä”
Author Response
Dear reviewer, thank you for your substantive comments and errors editing. We edited it in our text as you can see below.
- REVIEWER
1. Line 79 - the acronym ROS has already been introduced previously in the line 68, so there is no need to redefine it.
Text was edited.
- Line 96 - PCL acronym requires some explanation.
Text was edited.
- Line 122 - "Penta chemicals" the second part of the name has a lowercase letter, while on line 123 the uppercase letters have been used. Please standardize the spelling of the names of companies producing chemical reagents throughout the work.
Text was edited.
- Line 131 - why is “Scanning electron microscopy” capitalized?
Text was edited (recently line 152)
- Line 152 - notation of the unit 594 m3/h incorrect - the number 3 should be in the exponent.
Text was edited (recently line 172)
- Line 173 - the SI system of units applies, so the temperature should be written in Kelvin. Please standardize the entry throughout the work.
Units °C are commonly used and accepted in biological research articles.
- Line 207 - it should be "6 mm".
Text was edited (recently line 228).
- Line 213 - it should be “48h (D2), 72 h”.
Text was edited (recently line 234).
- Line 372 – it should be “Figure 7”
Text was edited, recently line 397.
- Figure 7 - What do the columns 1, 2, 3 mean in figure 7?
The explanation was put in a figure legend (line 429-430).
- Line 575 - it should be “[21]. Harri Hemilä”
We put correct citation number.
Best regards,
Eva Filova

Reviewer 2 Report
The authors investigated a novel functionalized nanofiber wound dressing for encapsulating antibacterial and antioxidant substances and applied it to a chronic wound model. The therapy results exhibit significant healing effects with low granulation, reepithelization, and inflammation. The work is comprehensive, from the material preparation and the therapy. I have some minor suggestions for your consideration.
1. In the article, the authors used a long description to describe why the addition of VitE did not positively affect the experimental results (lines 566-572). Since VitE did not have a good effect on the effect, why was it added to the experimental group? For comparison? What is the significance of this set of settings? Have you tried other substances?
2. As we can see from the marks of mousetails in Figures 10 and 11, the mouse of group PCL+Bac+Gen, PCL+Bac+Gen+VitE and Control you used in the figures were not the same ones. Can you explain why?
3. Line 429-432, something is wrong with the number of Figures 11, 12 and 13, which need to be confirmed. Meanwhile, the descriptions of Figure 13 were not found (Histological evaluation of newly formed tissue) and there is no interpretation of Figures 12 and 13.
4. Lines 434-438 could be put in Part 2 Material and Methods or supplementary information.
5. Figure 10 and Figure 11 could be combined into one figure as they all described the skin defects of mice. Identical to other Figures, each figure's formation and size is better to rearrange.
Author Response
Letter to reviewer
Dear reviewer, thank you for your substantive comments and errors editing. We edited it in our text as you can see below.
- REVIEWER
The authors investigated a novel functionalized nanofiber wound dressing for encapsulating antibacterial and antioxidant substances and applied it to a chronic wound model. The therapy results exhibit significant healing effects with low granulation, reepithelization, and inflammation. The work is comprehensive, from the material preparation and the therapy. I have some minor suggestions for your consideration.
- In the article, the authors used a long description to describe why the addition of VitE did not positively affect the experimental results (lines 566-572). Since VitE did not have a good effect on the effect, why was it added to the experimental group? For comparison? What is the significance of this set of settings? Have you tried other substances?
The prediction of the research was based on the research articles and the common knowledge was that VitE as an antioxidant combined with an antibiotic will reduce the wound area, healing time and quality of healing, so we add to the discussion all the results we get that showed preferences in the experimental group during the experiment.
In lines 599-606 we explained the hypothesis of SLP based on tetradecanol with VitE and all parameters prepared to release VitE into the wound and to obtain its expected positive effect on the wound healing process. Positive results from other publications are mentioned (citations 26, 27).
As we explained in the discussion, positive effect of VitE occurred in groups without bacteria therefore we proposed it as the second step of the therapy of chronic wounds after eradication of bacteria.
- As we can see from the marks of mousetails in Figures 10 and 11, the mouse of group PCL+Bac+Gen, PCL+Bac+Gen+VitE and Control you used in the figures were not the same ones. Can you explain why?
Results of in vivo experiments were presented from 50 mice and as numerous results were obtained from all experimental animals we chose animals with wounds that often occurred in the group and which represent each group well.
- Line 429-432, something is wrong with the number of Figures 11, 12 and 13, which need to be confirmed. Meanwhile, the descriptions of Figure 13 were not found (Histological evaluation of newly formed tissue) and there is no interpretation of Figures 12 and 13.
Figures 10 and 11 were put together and numbers of following figures numbers were changed. The description of Figure 12 was added. The Figure 8 was also modified and a new figure was added as a supplementum 1 according to another reviewer´s comment. The results were extended by the interpretation of results from histology (Figures 12-15).
- Lines 434-438 could be put in Part 2 Material and Methods or supplementary information.
The paragraph was cut and a basic information was left there. A graph of animal survival was added as a supplementum 1.
- Figure 10 and Figure 11 could be combined into one figure as they all described the skin defects of mice. Identical to other Figures, each figure's formation and size is better to rearrange.
Figures 10 and 11 were combined into one figure and numbers of the following figures were changed in the text. The Figure 8 was also modified.
Best regards,
Eva Filová

Reviewer 3 Report
The authors have presented an interesting research work on antibacterial and antioxidant substances loaded in a nanofiber and have done a detailed in-vitro and in-vivo study to demonstrate its efficacy to eliminate Staphylococcus aureus and Pseudomonas aeruginosa pathogenic bacterial cells. However there are some minor comments which the authors should address before the article could be accepted.
1. In figure 1, 3 and 4, the authors should mention in figure caption that these images are of SEM.
2. The author in 3.1.2 used the SLS technique for measuring the particle size. It would be interesting to know why the author did chose SLS and not the standard DLS (dynamic light scattering)?
3. Could the authors evaluate the poly-dispersity index for their nanofibers using SLS techcique?
4. There are several biocompatible nanomaterials developed which have potent antimicrobial activity against multidrug resistant pathogenic bacterial strains. The authors should add few of such studies in their introduction or discussion such as (Synthesis of low molecular weight alginic acid nanoparticles through persulfate treatment as effective drug delivery system to manage drug resistant bacteria) and (Amelioration Studies on Optimization of Low Molecular Weight Chitosan Nanoparticle Preparation, Characterization With Potassium Per Sulphate and Silver Nitrate Combined Action With Aid of Drug Delivery to Tetracycline Resistant Bacteria).
5. The authors could do a statistical analysis of data in graphs of figure 6.
6. Could the authors include a survivability data (Kaplan–Meier plot) based on the in-vivo activity of the wound healing of the nanofiber?
Author Response
Letter to reviewers
Dear reviewer, thank you for your substantive comments and errors editing. We edited it in our text as you can see below.
- REVIEWER
The authors have presented an interesting research work on antibacterial and antioxidant substances loaded in a nanofiber and have done a detailed in-vitro and in-vivo study to demonstrate its efficacy to eliminate Staphylococcus aureus and Pseudomonas aeruginosa pathogenic bacterial cells. However there are some minor comments which the authors should address before the article could be accepted.
- In figure 1, 3 and 4, the authors should mention in figure caption that these images are of SEM.
The information was added.
- The author in 3.1.2 used the SLS technique for measuring the particle size. It would be interesting to know why the author did choose SLS and not the standard DLS (dynamic light scattering)?
Based on our experience the size 100 µm is not suitable for DLS method. The SLS method gave us good results about the particle size of the dispersion.
- Could the authors evaluate the poly-dispersity index for their nanofibers using SLS technique?
I think it is impossible to get the nanofiber as units from the substrate and put them in a liquid environment and keep them stable during the measurement. The SLS method uses data correlation calculating diameters to spherical particles that also do not fit into fibers. There are various methods to evaluate it using SEM and others, but the more common one is to use the average fiber diameter. Therefore, we cannot use the same method to obtain the polydispersity of our nanofibers in the substrate.
- There are several biocompatible nanomaterials developed which have potent antimicrobial activity against multidrug resistant pathogenic bacterial strains. The authors should add few of such studies in their introduction or discussion such as (Synthesis of low molecular weight alginic acid nanoparticles through persulfate treatment as effective drug delivery system to manage drug resistant bacteria) and (Amelioration Studies on Optimization of Low Molecular Weight Chitosan Nanoparticle Preparation, Characterization With Potassium Per Sulphate and Silver Nitrate Combined Action With Aid of Drug Delivery to Tetracycline Resistant Bacteria).
We added few relevant studies in the introduction (Lines 104-127). Two recommended articles are not relevant to our research or to comparison of the results.
- The authors could do a statistical analysis of data in graphs of figure 6.
We added the information into Methods –lines 221-222 and Figure 6 caption.
For statistical evaluation SigmaStat 3.5 software and a One Way ANOVA test were used.
Figure 6 caption
The statistical analysis shows there is not statistically significant difference for STAU (P=0.152) or for PSAE (P=0.164).
- Could the authors include a survivability data (Kaplan–Meier plot) based on the in-vivo activity of the wound healing of the nanofiber?
Tha Kaplan-Meier plot was added as a Supplementum 1.
Best regards,
Eva Filová
